# Metagenomic and Physicochemical Analyses Reveal Microbial Community and Functional Differences Between Three Different Grades of Hongxin Low-Temperature Daqu

**DOI:** 10.3390/foods14071104

**Published:** 2025-03-22

**Authors:** Chao Ren, Mengke Zhao, Tinghui Xue, Tianpei Geng, Xiao Nie, Chaoyue Han, Yuge Wen, Liyan Jia

**Affiliations:** 1College of Food Science and Engineering, Shanxi Agricultural University, Jinzhong 030801, China; renchao2333@outlook.com (C.R.); z18339058292@163.com (M.Z.); xuetinghui569@163.com (T.X.); gziyan2023@126.com (T.G.); 13852163371@163.com (X.N.); 19726869037@163.com (C.H.); m13294653450@163.com (Y.W.); 2Graduate Education Innovation Center on Baijiu Bioengineering in Shanxi Province, Jinzhong 030801, China; 3Industry Technology Innovation Strategic Alliance on Huangjiu in Shanxi Province, Jinzhong 030801, China

**Keywords:** Hongxin low-temperature Daqu, metagenomic sequencing, physicochemical properties, microbial diversity and function

## Abstract

Hongxin (HX) is an indispensable Daqu in the production of light-flavor Baijiu (LFB). However, the classification method of HX is highly subjective, and the classification and functional differences in microorganisms in different grades of HX are still unclear. In this study, metagenomics and physiochemical analysis were used to compare three grades of HX (top, first, second) and clarify their brewing functions in LFB. The results showed that a total of 1556 genera and 5367 species were detected in all samples. Bacteria and fungi are the main microorganisms in HX, and the relative abundance of bacteria and fungi is above 4.5:1. *Kroppenstedtia* (11.43%), *Leuconostoc* (10.52%), *Fructilactobacillus* (9.00%) were the top three genera in HX. Although the microbial community composition of the three grades of HX is highly similar, each HX has a specific microbial community structure and macrogene functional characteristics, indicating that they have different brewing functions. The dominant microorganisms in top-grade HX and first-grade HX were mainly positively correlated with energy metabolism and lipid metabolism, while the dominant microorganisms in second-grade HX were mainly positively correlated with carbohydrate metabolism and amino acid metabolism. This study revealed the different fermentation effects of different grades of HX in LFB and provided suggestions for the scientific classification and quality control of HX.

## 1. Introduction

Chinese Baijiu is one of the oldest distilled spirits in the world, with a long historical legacy and deep cultural connotations [1]. Factors such as raw materials, brew type, brewing process, and natural environment determine the style of Baijiu [2]. Light-flavor Baijiu (LFB) is one of the three representative Baijiu in China [3], and it is considered to be the origin of Chinese Baijiu, with a market share of about 75% at one time [4]. Characterized by its pure flavor—mellow and sweet—natural coordination, and clean aftertaste, it is well received by consumers [5]. The fermentation of LFB is a unique and complex process, with simultaneous saccharification and natural fermentation using low-temperature Daqu (LTD) as the fermenting agent and fermentation and distillation under solid-state conditions [6]. The LTD includes three types of Daqu, namely Qingcha (QC), Hongxin (HX), and Houhuo (HH) [7]. The types of LTD under discussion employ the same set of ingredients for manufacturing (a 6:4 ratio between barley and peas) [8]. Their production methodologies exhibit considerable similarity. What sets them apart is their unique temperature specifications during both the fermentation and drying phases, which are tailored to individual requirements for each type of LTD [9]. The production of HX requires more stringent conditions. Compared with the other two types of Daqu, HX requires raising the temperature of the Daqu to its highest level and maintaining it for 4 to 5 days [10]. Varying fermentation temperatures gives rise to distinctive microbiological growth dynamics and fermentation traits intrinsic to different LTD, thereby engendering disparities in their perceptual qualities, operational functionalities, and resultant metabolites.

Since Daqu is prepared under semi-controlled conditions, the quality of Daqu varies [11]. Producers grade Daqu to reduce the instability in the Baijiu production process caused by differences in Daqu quality [12]. However, the method of grading Daqu based on its color and odor is highly subjective. Taking HX as an example, HX is classified into three grades, namely top grade, first grade, and second grade, according to the characteristics and odor of its central part. Top-grade HX (HXT) is characterized by a distinct red line in the middle and has the aroma of roasted peas [13]. First-grade HX (HXF) has a faint red line or cracks, and its odor is not typical. Second-grade HX (HXS) has no red line and gives off an acidic or foul odor. However, it is difficult to explain the quality attributes of Daqu using only sensory evaluation terms. This highlights the necessity of studying the microbiota and quality attributes of HX at different grades to standardize the production process and ensure the quality of Baijiu.

Early studies of the microbial community in Daqu were dominated by the traditional isolation and culture method [14,15,16], which although valuable in culturing and characterizing the microbial community in Daqu, can only isolate, cultivate, and identify culturable microorganisms, which significantly underestimates the microbial diversity in the Daqu samples. In recent years, next-generation sequencing technologies have enabled high-throughput sequencing analyses, increased sample throughput, higher sensitivity, and greater data generation, and it allowed for the detection of low-abundance species [17], dramatically changing how microbial communities can be detected in Daqu. However, this technique is limited to genus-level identification [18] and may produce inaccurate results [19]. Therefore, there is a need for more reliable methods to detect microbial diversity in sake at the species level, and metagenomic shotgun sequencing accurately characterizes taxonomic composition and generates functional profiles through the direct gene identification of the microbiome [20]. In addition, two or more conditions associated with Daqu can be assessed by comparing metagenomic analysis. Based on this analysis, it is possible to accurately and efficiently identify genes with abundant differences among metagenomes and obtain a series of rich information on the structure, diversity, pathway subsystems, and biological function differences in microbial communities [21]. The method is now widely used to analyze a variety of microbial ecosystems, including those found in food and the human body [22,23,24]. However, it is rarely used to characterize the microbiota and functional characteristics of HX at different grades.

In this study, we investigated different grades of HX using metagenomics and conducted a comparative study on the structure and functional characteristics of microbial communities of different grades of HX. The physicochemical and enzymatic indexes of various grades of HX were also analyzed, including moisture, acidity, saccharification power, liquefaction power, esterification power, alcoholic power, and fermentation power, which were very important in revealing the quality differences between them. This study mainly clarified the characteristics, functions, and quality characteristics of microbial communities in different grades of HX, further revealed the role of different grades of HX in the production of LFB, and provided valuable insights for improving the quality evaluation method and quality of HX.

## 2. Materials and Methods

### 2.1. Sample Collection

The HX samples were made from barley (60%) and peas (40%), and they were obtained from an LFB distillery in Fenyang, Shanxi Province, China. All HX samples underwent a complete fermentation and storage process. The three grades of HX (HXT, HXF, and HXS) were determined by experienced workers based on the appearance, aroma, and fracture surface color (Figure 1). A total of 9 Daqu bricks were screened out, and then they were crushed and mixed to obtain 3 representative samples. The Daqu powder was divided into two parts, transferred to sterile packaging, and then transported to the laboratory through the cold chain. One part was used to determine the subsequent physical and chemical parameters (stored at 4 °C), and the other part was used to analyze the microbial community (stored at −80 °C).

### 2.2. Determination of Physicochemical and Enzymatic Properties

The physical and chemical indexes of HX were determined in triplicate. According to the determination method specified in the industry standard QB/T 4257-2011 [25], five parameters were measured to evaluate the standard enzyme activity, namely saccharification power, liquefaction power, fermentation power, esterification power, and alcohol power. In addition, two physical and chemical parameters of moisture and acidity were determined. The gravimetric method determined moisture (105 °C, 4 h). Acidity was determined by titration with NaOH (0.1 mol/L) and phenolphthalein (endpoint pH: 8.2) [6].

### 2.3. Sample DNA Extraction, Library Preparation, and Data Quality Control

DNA extraction was performed using MagPure Stool DNA KF Kit B (Magen, Guangzhou, China). An amount of 100–200 mg of HX samples were lysed with Buffer ATL (QIAGEN, Hilden, Germany), ground, centrifuged, and the supernatant was treated with Buffer PCI. The magnetic bead binding solution (containing protease K and RNase A) was purified and separated by Kingfisher automated system to finally obtain DNA. The library was constructed using the MGIEasy DNA Library Prep Kit (BGI, Shenzhen, China). After DNA fragmentation, magnetic bead screening, end repair plus A, adaptor ligation, and PCR amplification, the qualified library was subjected to denaturation and cyclization into single-stranded DNA nanospheres (DNB). The phi29 polymerase rolling circle amplification was used to perform PE150 sequencing on the DNBSEQ-G400 platform by cPAS technology. The original data were filtered and quality-controlled using SOAPnuke (2.3) software [26]. The command was ‘SOAPnuke filter-20-q 0.5-n 0.01-d-Q 2-50-adaMis 0.3’, and then Bowtie2 (2.4.4) [27] was used to compare the host sequence (human and plant) and remove the sequence in the ratio to generate clean data. The detailed sequencing statistics are shown in Appendix A.

### 2.4. Data and Statistical Analysis

Firstly, MEGAHIT (version 1.2.9) was used to assemble the quality control and dehosted sequences based on k-mer to generate contigs, and then MetaGeneMark-2 software was used to predict the gene sequence in contigs. CD-HIT (version 4.6.8)software was used to remove the redundancy of the obtained genes, Salmon (version 1.10.1)software was used to calculate the relative abundance table of each gene, and then RGI (version 6.0.2) was used to compare the non-redundant genes to eggNOG, KEGG, and CAZy databases to complete gene function annotation. Kraken2 and a self-built database (screening NCBI NT database) were used to calculate the number of sequences of the species contained in the sample, and then Bracken2 was used to estimate the actual abundance of the species in the sample to complete the species annotation. Then, based on the gene abundance table, species abundance table, and functional abundance table, the distribution of genes, species, and functions was visualized.

Stacked histograms, bar charts, and boxplots were displayed by GraphPad Prism 9.0. Statistical significance and the Spearman correlation coefficient were calculated by SPSS 24.0 (IBM Corporation, Armonk, NY, USA). STAMP (version 2.1.3) software was used to analyze the differences in the CAZY family among Daqu samples. RStudio (version 4.2.2) software was used for Venn diagrams, principal component analysis (PCA), linear discriminant analysis (LefSe), and heat maps. Canoco 5 software was used for redundancy analysis (RDA).

## 3. Results

### 3.1. Microbial Composition Based on Shotgun Metagenomics

Metagenomic sequencing was performed on nine HX samples. A total of 93.94 Gbps of raw data were generated in this study. The Q20 (sequencing error rate < 1%) and Q30 (sequencing error rate < 0.1%) of all samples was greater than 97% and 93%, respectively, indicating high sequencing quality and reliable data. After strict quality control and the removal of host DNA readings, each sample produced an average of 8.32 gigabase (Gb) double-ended read length, totaling 74.92 Gb of high-quality data (Appendix A).

At the boundary level, HX was composed of bacteria (81.98%), eukaryotes (17.98%), archaea (0.04%), and viruses (0.01%). Bacteria and eukaryotes are the main microflora of HX, and bacteria are dominant. A total of 58 phyla, 1554 genera, and 5363 species (Appendix A) were identified. At the phylum level, the average relative abundance above 1% (Figure 2a) corresponded to Bacillota (68.68%), Ascomycota (16.33%), Actinomycetota (9.25%), Pseudomonadota (3.89%), and Basidiomycota (1.09%). At the genus level, there are 16 genera, with an average relative abundance of more than 1% (Figure 2b). They were *Kroppenstedtia* (11.43%), *Leuconostoc* (10.52%), *Fructilactobacillus* (9.01%), *Bacillus* (8.93%), *Pichia* (7.86%), *Thermoactinomyces* (7.42%), *Aspergillus* (5.73%), *Lactiplantibacillus* (4.86%), *Streptomyces* (4.39%), *Companilactobacillus* (4.32%), *Acetobacter* (1.84%), *Latilactobacillus* (1.83%), *Pediococcus* (1.78%), *Levilactobacillus* (1.41%), *Saccharopolyspora* (1.31%), and *Staphylococcus* (1.2%). At the species level, there were 16 species with an average relative abundance of more than 1% (Figure 2c), namely *Kroppenstedtia eburnea* (11.30%), *Fructilactobacillus sanfranciscensis* (9.00%), *Leuconostoc citreum* (8.36%), *Pichia kudriavzevii* (7.86%), *Thermoactinomyces vulgaris* (7.42%), *Bacillus licheniformis* (6.52%), *Lactiplantibacillus plantarum* (4.44%), *Aspergillus oryzae* (3.00%), *Streptomyces albus* (2.83%), *Companilactobacillus paralimentarius* (2.59%), *Pediococcus pentosaceus* (1.55%), *Levilactobacillus brevis* (1.40%), *Companilactobacillus crustorum* (1.30%), *Aspergillus flavus* (1.28%), *Latilactobacillus sakei* (1.23%), and *Acetobacter pasteurianus* (1.04%). It is worth noting that some microorganisms with low relative abundance (<1%) that play an important role in Baijiu fermentation, such as *Saccharomyces*, *Talaromyces*, and *Kluyveromyces*, were also detected in HX, with relative abundances of 0.67%, 0.16%, and 0.08%, respectively. Due to the use of the same raw materials and similar environmental conditions in the production of these three grades of HX, the microbial communities in these three grades of HX showed a lot of overlap. Nevertheless, the main differences in the three grades of HX are caused by small differences in the fermentation process.The Venn diagram (Figure 2d,e) illustrates the common and unique species of three grades of HX and nine HX samples, respectively. Among the 5363 species identified in this study, 2455 species could be detected in all three grades of HX. It should be noted that each sample contained 1446 core species, and the cumulative proportion of these species in each sample reached 95.78% (sd = 1.88%) of the total microorganisms on average, which means that all three grades of HX samples have a considerable number of species shared and are common in all individual samples. In addition, 355, 432, and 642 species were unique to HXT, HXF, and HXS, respectively, accounting for only 0.15%, 1.14%, and 1.54% of the relative abundance of their microbial communities. These findings suggest that only a small number of microorganisms with a very low proportion of relative abundance are particularly associated with a specific grade of HX.

### 3.2. Differences in the Microbial Community Structure Among the Three Grades of HX

To fully understand the microbial diversity and richness of the three grades of HX, the species-level α-diversity of the three grades of HX was measured and compared by Chao1, Shannon, and Simpson indices (Figure 3a–c). There was no significant difference in the diversity index of the three grades of HX, but it is worth noting that the diversity index of HXS was the lowest. Principal coordinate analysis (based on Bray–Curtis and Jensen–Shannon divergence distance) was used to further explore the differentiation of microbial community structure in three grades of HX type (Figure 3d,e). The results show that the distance between the samples of the three grades of HX is far. This indicates that there are indeed significant differences between the three grades of HX-related microbial communities. The LEfSe algorithm was used to identify enrichment features for specific HX samples, and only taxa with average relative abundances above 0.5% were considered for analysis (Figure 3f,g). HXF had the highest number of significantly enriched taxa, followed by HXS, while HXT was the least enriched. At the phylumlevel, Bacillota was significantly enriched in all three grades of HX, Actinomycetota in HXF, and Pseudomonadota and Ascomycota in HXT and HXF. At the family level, Lactobacillaceae and Erwiniaceae were significantly enriched in HXT; Streptomycetaceae, Lactobacillaceae, Acetobacteraceae, Aspergillaceae, and Saccharomycetaceae were significantly enriched in HXF; and Thermoactinomycetaceae and Lactobacillaceae were significantly enriched in HXS. At the genus level, *Lacticaseibacillus*, *Leuconostoc*, and *Pantoea* were significantly enriched in HXT; *Streptomyces*, *Fructilactobacillus*, *Lacticaseibacillus*, *Leuconostoc*, *Acetobacter*, *Aspergillus*, and *Saccharomyces* were significantly enriched in HXF; and *Kroppenstedtia*, *Thermoactinomyces*, *Companilactobacillus*, *Leuconostoc*, *Levilactobacillus*, and *Pediococcus* were significantly enriched in HXS. Species significantly enriched in HXT were *Lacticaseibacillus paracasei*, *Leuconostoc citreum*, and *Pantoea agglomerans*. Species significantly enriched in HXF were *Streptomyces sp NHF165*, *Fructilactobacillus sanfranciscensis*, *Latilactobacillus curvatus*, *Latilactobacillus sakei*, *Leuconostoc mesenteroides*, *Acetobacter pasteurianus*, *Aspergillus nidulans*, and *Saccharomyces cerevisiae*, while the species significantly enriched in HXS were *Kroppenstedtia eburnea*, *Thermoactinomyces vulgaris*, *Companilactobacillus crustorum*, *Leuconostoc pseudomesenteroides*, *Levilactobacillus brevis*, and *Pediococcus pentosaceus* (all belonging to Bacillota).

### 3.3. Metagenomic Functional Characteristics in Three Grades of HX

The metagenomic data were annotated using the KEGG, CAZy, and eggNOG databases to explore the differences in functional characteristics and metabolic pathways among the three grades of HX. At level 1 of the KEGG database (Figure 4a), the abundance of metabolism-related genes in all HX flora was the most abundant, accounting for 74.01%, 73.71%, and 78.48% of the annotated genes of HXT, HXF, and HXS, respectively. Therefore, a further analysis of KEGG pathway level 2 (Figure 4b), where carbohydrate metabolism and amino acid metabolism are the most dominant biological functions, suggests that Daqu has a great potential for substrate degradation and flavor formation. Considering the highest abundance of carbohydrate metabolism in the Daqu flora and the highest starch content in the raw material, it is important to reveal the differences in carbohydrate utilization capacity among the three Daqu species based on the CAZy database. Among the six functional categories annotated, glycosyl hydrolases (GHs) and glycosyltransferases (GTs) were dominant in all samples (Figure 4c). Among them, the abundance of GHs and GTs was higher in HXT and HXS than in HXF, while the abundance of CBMs was significantly highest in HXS. In addition, the annotation results based on the eggNOG database showed that the three grades of HX had the largest number of genes for amino acid transport and metabolism (E: 9435) and carbohydrate transport and metabolism (G: 8580), except for unknown function (S: 30508) and transcription (K: 9236) (Figure 4d). These results are consistent with the functional spectrum obtained from the KEGG database, indicating that the synthesis and metabolism of amino acids and carbohydrates are the most important biological functions of HX.

When comparing the microbial carbohydrate metabolism genes of the three classes of HX (Figure 4e), it was observed that nine CAZy families (CBM50, GT4, GT119, GT51, GH23, GH28, GH0, GH3, and CE4) had the highest relative abundance in HXS; six CAZy families (AA6, GT49, GT0, GH47, GH5, and GH152) had the highest relative abundance in HXT; CMB13, GT35, GT22, GH73, and GH34 had the highest relative abundance in HXF. STAMP analysis was then used to screen for differences in the relative abundance of the three classes of HX CAZy families (*p* < 0.05), reflecting the carbohydrate-utilizing capacity of Daqu (Figure 4f). The relative abundance of the HXS versus the HXF CAZy families differed significantly. Intergroup difference analysis revealed that a total of 20 CAZy families differed significantly in relative abundance, with the relative abundances of GT51, GH170, GT119, CBM50, GH28, GH0, and GT4 being significantly higher in HXS than in HXF. The intergroup difference analysis between HXS and HXT revealed that a total of 18 CAZy families differed significantly in relative abundance, with the relative abundances of GH0 in HXS, GH170, GH28, CBM50, GT51, GH23, GT119, and GT4 being significantly higher than those of HXF. Appendix A provides detailed information on the functional characteristics represented by these CAZy families. In contrast, there was no significant difference in the relative abundance of CAZy families between HXT and HXF.

### 3.4. Correlation Between the Microbial Group of HX and the Physical and Chemical Properties and the Function of Metagenes

Physicochemical and enzymatic indicators reflect the state and quality of Daqu fermentation. Histograms were used to visualize the content of two physicochemical parameters and the activity of five enzymes for the three HX samples (Figure 5a). The moisture of the HX samples was below 15%. In terms of the three parameters of acidity, saccharification power, and liquefaction power, the phenomenon that HXT is higher than HXF, which is in turn higher than HXS, was observed. Additionally, significant differences were found among HXT, HXF, and HXS (*p* < 0.01). HXS had a significantly higher moisture and alcohol power than the other two Daqu (*p* < 0.05); it is worth noting that HXS had a significantly lower fermentation power and esterification power than the other two Daqu (*p* < 0.05).

To further infer the effects of microbial enzyme production and abiotic factors on different HX microorganisms, redundancy analysis (RDA) was performed on abiotic factors, enzyme activities, and different Daqu microorganisms (Figure 5b). Moisture affected HXS more significantly than the other two Daqu, while HXT was more affected by acidity. There was a positive correlation between fermentation power, esterification power, acidity, liquefaction power, and saccharification power versus a negative correlation between alcoholic power and moisture. Then, correlation heat maps (based on Spearman correlation coefficients for rank correlations of dominant species detected in HX samples) were constructed to visualize the associations between microbiota and physicochemical–enzymatic properties (Figure 5c) and metagenomic functions (Figure 5d) in HX. Representative dominant microorganisms in HXT such as *Leuconostoc citreum* were positively correlated with esterification power, liquefaction power, and acidity and negatively correlated with liquefaction power and moisture. In addition, *Fructilactobacillus sanfranciscensis*, *Saccharomyces cerevisiae*, *Latilactobacillus sakei*, *Aspergillus nidulans*, and *Saccharopolyspora rosea* were positively correlated with the fermentation power; *Companilactobacillus paralimentarius* was positively correlated with esterification power; and *Kroppenstedtia eburnea*, the species with the highest relative abundance in HXS, was negatively correlated with fermentation power, which may be the reason why the fermentation power of HXS was lower than that of the other two HX. Three groups were identified in the cluster analysis of the correlation between dominant microbial species and metagenomic functions. The representative species in group (1), group (2), and group (3) accounted for the highest relative abundance of the microbial flora in HXT, HXF, and HXS, respectively. At KEGG level 2, there is a positive correlation between group (1) and group (2) in aspects such as lipid metabolism, energy metabolism, and nucleotide metabolism. On the contrary, there is a negative correlation in aspects like carbohydrate metabolism, amino acid metabolism, and the metabolism of terpenoids and polyketides. Compared with the dominant species in HXT and HXF, the representative dominant species group (3) in HXS showed a completely opposite trend in the relationship of metagenomic function. These findings suggest that the microbiota in HXS, HXT, and HXF may be functionally different during the LFB brewing process.

## 4. Discussion

HX is one of the starter cultures of LFB. The quality and yield of different grades of HX have an important influence on the flavor and quality of LFB. In this study, the metagenome and physicochemical properties of three different grades of HX were compared and analyzed, and the quality differences between different grades of HX were comprehensively and scientifically evaluated, which provided a reference for improving the production process.

First, the microorganisms of HX were analyzed based on macrogenomes, which showed that a total of 1556 genera and 5367 species were detected in all HX samples. Since all three grades of HX were produced using the same raw materials, they had a large overlap of core microbial sets. However, significant differences were still detected by LEfSe analysis. In addition, it was shown that HX was mainly composed of bacteria and fungi with relative abundances of 81.98% and 17.98%, respectively. In addition, the study found that with the decrease in HX grade, the proportion of fungi decreased, which may be the result of the difference in fermentation temperature. *Kroppenstedtia eburnea* is the most abundant microorganism in HXS (32.96%), while *K.eburnea* is generally the dominant bacteria in high-temperature Daqu [28]. This indicates that the temperature of HXS may fluctuate too much during the fermentation process, and the maximum temperature exceeds the maximum temperature of HX production by 47 °C, while most fungi are not heat-resistant and will be inhibited under high temperature conditions [8].

The quality difference between different grades of HX is still the focus of research. In this study, three grades of HX were comprehensively compared, and the microbial community, function, and physicochemical properties of HX were discussed. Although there was no significant difference in the microbial diversity at the species level among the three grades of HX, compared with HXT and HXF, HXS showed the lowest α-diversity, which indicates the presence of certain dominant microbial communities with relatively high abundances in HXS. The Chao1 index, which represents the species richness index, and the Shannon and Simpson diversity indexes, which measure species diversity and evenness, are the highest in HXT. These results indicate that the microbial community in HXT is more abundant and diverse. This may be due to the fact that the temperature in the production process is within the optimal temperature range of HX, which is conducive to the growth of microorganisms with low heat resistance. At the same time, β-diversity analysis confirmed that different grades of HX samples did show different microbial community structures. The HXS flora is dominated by *Kroppenstedtia eburnean* (32.96%) and *Thermoactinomyces vulgaris* (8.12%). These two microorganisms tend to be found in medium- and high-temperature Daqu [29,30,31], which side-steps the possibility that the fermentation temperature of HXS might be too high. *K. eburnea* can produce α-amylase to provide the liquefaction capacity of Daqu [28], but the liquefaction power of HXS is the lowest among the three Daqu, and it is possible that other microorganisms or environmental factors also influence the liquefaction power [32].

*Thermoactinomyces vulgaris* is a core functional bacterial microbiota in Daqu and fermented grains [33]. It has a relatively high abundance in all three types of HX (with an average relative abundance of 7.42%). It has typical thermophilic characteristics and often appears in high-temperature fermentation sites such as compost, straw, and bagasse. It not only secretes proteases, cellulases, pectinases, lysozymes, thermophilic amylases, and neobranched amylases but also produces volatile compounds such as pyrazine, which provides a spicy flavor, and geosmin, which provides an earthy flavor. *Pichia kudriavzevii* (12.62%) and *Fructilactobacillus sanfranciscensis* (17.64%) were the most abundant microorganisms in HXT and HXF. *P. kudriavzevii* can degrade organic acids and release various hydrolases [31], which can improve the fermentation ability of Daqu [32,34]. *F. sanfranciscensis* metabolizes carbohydrates such as sucrose or maltose to produce extracellular polysaccharides (EPSs) and hydrolyzes proteins into peptides [35].

It is noteworthy that the average relative abundance of *Leuconostoc citreum*, *Bacillus licheniformis*, and *Aspergillus* in these three classes of HX can reach 8.36%, 6.52%, and 5.73%, respectively. *L. citreum* is a major functional microorganism in the synthesis of Daqu flavor compounds, which is involved in the synthesis of Daqu amino acids and plays an important role in the synthesis of higher alcohols, esters, and other flavor compounds in Baijiu [36]. At the same time, it is also involved in the synthesis of organic acids in Daqu [37], which plays a role in eliminating bitterness, reducing astringency, buffering taste, and stabilizing wine aroma [38]. *Bacillus licheniformis* is an important functional strain in Baijiu fermentation, which can produce more than 70 metabolites with significant effects on aromatic compounds in Daqu [39], as well as carbohydrases and proteases, which can effectively improve the efficiency of Baijiu fermentation and play an important role in the production of various flavor compounds [5]. *Aspergillus* was positively correlated with the fermentation activity of Daqu, as well as with the synthesis of volatile organic compounds in Daqu, such as ethyl pentadecanoate and ethyl myristate [40].

A number of low-relative-abundance microorganisms that positively affect Baijiu fermentation were also detected in HX, such as *Saccharomyces*, *Talaromyces*, and *Kluyveromyces*. *Saccharomyces* proved to be a major contributor to the production of higher alcohols (isobutanol, isoamyl alcohol, and 1-propanol) and had a significant impact on volatile metabolites [41]. *Talaromyces*, an active functional microorganism producing volatile flavors in white wine fermentation, is positively correlated with the production of ethyl hydroxyhexanoate and diethyl succinate [42]. *Kluyveromyces* can produce many enzymes, such as β-galactosidase, lipase, protein phosphatase, carboxypeptidase, and β-glucosidase. Therefore, it can catalyze the conversion of macromolecules to small molecules and the biological production of flavor compounds [43]. Therefore, the function of microorganisms with low relative abundance is not negligible and is worth studying.

The KEGG and eggNOG databases were utilized to annotate the macrogenomic data of HX. The highest number of genes related to carbohydrate metabolism and amino acid metabolism was found in three classes of HX in the KEGG and eggNOG databases, which is consistent with previous studies [44]. Subsequently, the CAZy database was used to classify and annotate the modules related to carbohydrate-active enzymes in HX-related microbial metagenomic data. Among the six main functional module categories, GH and GT had the highest content, while PL had the lowest abundance [45]. Glycoside hydrolases are involved in the degradation of polysaccharides such as starch, glucose, and cellulose, and they decompose the grain raw materials in Daqu into key resources available to microorganisms [38]. Glycosyltransferases can affect the functions and properties of proteins, lipids, and nucleic acids by connecting different types of sugar groups, forming polysaccharide chains, or modifying pre-existing sugar chains [46]. Polysaccharide lyases act on anionic polysaccharides mainly through the β-elimination mechanism on the (1,4) glycosidic bond, resulting in the cleavage of anionic polysaccharides into oligosaccharides, especially in the case of polymers such as alginate or glucuronic acid that are only depolymerized by these polysaccharide lyases [47]. This indicates that the ability of microorganisms in HX to degrade and utilize polysaccharides such as starch, glucose, and cellulose is strong, but the ability to lyse anionic polysaccharides (such as gum and glycosaminoglycans) is weak. Among the three HXs, HXS has the highest abundance of glycoside hydrolase and carbohydrate-binding modules, which represents that HXS has a strong ability to degrade polysaccharides such as starch and glucose. At the same time, it can also enhance enzymatic hydrolysis and increase the proximity of the enzyme to its substrate, especially for insoluble substrates, thereby improving the efficiency of polysaccharide degradation [48]. In addition, no significant difference was found in the relative abundance of the CAZy family between HXT and HXF. This may be because the fermentation temperature of HXF fluctuates less and is close to the optimal temperature of red yeast rice, resulting in a relative abundance of microbial populations comparable to that of HXT. The secondary pathways of CAZy, GH34, GT4, CE4, GH13, GT2, and CBM50 all have relatively high abundances in HX of all grades. Among them, GT2 (glycosyltransferase 2) is involved in the synthesis of polysaccharides in the bacterial cell wall (such as peptidoglycan), and it may promote the production of lactic acid metabolites [49]. GH13 (glycoside hydrolase 13) belongs to the α-amylase family, which dominates the degradation of starch into maltose and glucose and is a core enzyme system for the saccharification function [50]. The high abundance of these families reflects the common metabolic demands of HX for substrates such as starch and cellulose and is consistent with the functional preferences of core microbial communities such as *Lactobacillus plantarum* [51], *Bacillus* [52], and *Aspergillus* [53].

The physicochemical results showed that although the different grades of HX were prepared from the same raw materials, they all exhibited significant differences in their physicochemical indexes. The moisture of HXS was the highest among the three types, which was probably due to the fact that it was in the inner layer of the Daqu brick fermentation pile, where the airflow was small and retained more moisture. In addition, in the three indexes of saccharification power, liquefaction power, and esterification power, all of them showed the phenomenon of decreasing with the decrease in HX grade, while the opposite phenomenon was observed in the index of alcoholic power. A correlation analysis showed that the saccharification power, liquefaction power, and esterification power of HX were positively correlated with *Companilactobacillus paralimentarius*, *Pantoea agglomerans*, and *Leuconostoc citreum* and negatively correlated with *Weissella cibaria*. The alcoholic power was positively correlated with *W.cibaria* and negatively correlated with *C.paralimentarius*, *P.agglomerans,* and *L.citreum*. Interestingly, the relative abundance of *C.paralimentarius*, *P.agglomerans,* and *L.citreum* decreased with the decrease in grade, while the relative abundance of *W.cibaria* increased with the decrease in grade, indicating that the physicochemical indexes of HX may be related to its microbial composition. In addition, the correlation between moisture and dominant microorganisms was almost the same as that of alcoholic power. (except *Bacillus licheniformis* and *Acetobacter pasteurianus*). The moisture difference in different HX will affect the microbial composition, and the microorganisms will affect the quality index of HX.

In addition, the dominant microorganisms in HXS are positively correlated with carbohydrate and amino acid metabolism. This indicates that HXS has better saccharification efficiency, the ability to produce alcohol, and the ability to utilize nitrogen sources, which is consistent with the results of the analysis based on the CAZy database. In contrast, the representative dominant microorganisms in HXT and HXF are positively correlated with energy metabolism and lipid metabolism, suggesting that they are more involved in the synthesis of esters and have high-efficiency energy conversion capabilities. These results show that the grade differences in HX are not only related to abiotic factors (moisture, acidity) and enzyme activities but are more closely related to the differentiation of the metabolic functions of the microbial community.

In summary, although a large proportion of the microbial communities in the three grades of HX overlaps, it is still possible to identify group-specific microbiome characteristics and representative dominant microorganisms. Some similarities were observed between the representative dominant microorganisms in HXT and those in HXF, while the functional metagenomic analysis of the representative dominant microorganisms in them and those in HXS indicates that they are highly different during the Baijiu fermentation process. These findings provide valuable insights into the physicochemical enzymological properties, microbial composition, and functions of HXS. Notably, based on the water content determination results and the microbial community composition of HXS, it is speculated that the main reason for the formation of HXS is the differences in moisture and temperature compared with HXT and HXF. These findings suggest that in the actual production of Daqu, a multi-dimensional temperature and humidity detection system can be established to better establish the correspondence between the grade of Daqu and the temperature and humidity in the fermentation chamber so as to avoid an uneven fermentation of Daqu (such as the accumulation of fermentation heat inside the Daqu pile), which affects the quality of Daqu. This contributes to the establishment of a scientific and refined Daqu quality control system, thereby improving the quality and production efficiency of LFB.

## 5. Conclusions

This study comparatively analyzed the physicochemical properties and metagenomics of three different grades of HX, uncovering the microbial community characteristics, functions, and quality attributes within HX of various grades. The results indicate that the proportion of fungi and bacteria declines as the grade of HX decreases. Although there is a significant overlap in the microbial species among the three grades of HX, the microbiota of each HX possesses specific microbial community and functional metagenomic features, suggesting that they perform distinct functions during the LFB fermentation process. This research enhances our understanding of different graded HXs and offers suggestions for establishing a scientific Daqu quality control system and promoting the development of the Baijiu industry.

## Figures and Tables

**Figure 1 foods-14-01104-f001:**
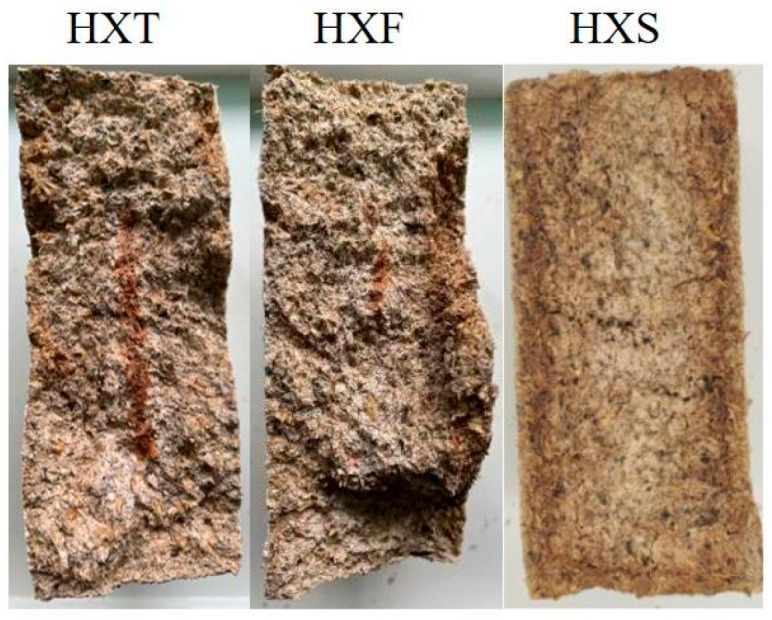
Cross sections of three grades of HX.

**Figure 2 foods-14-01104-f002:**
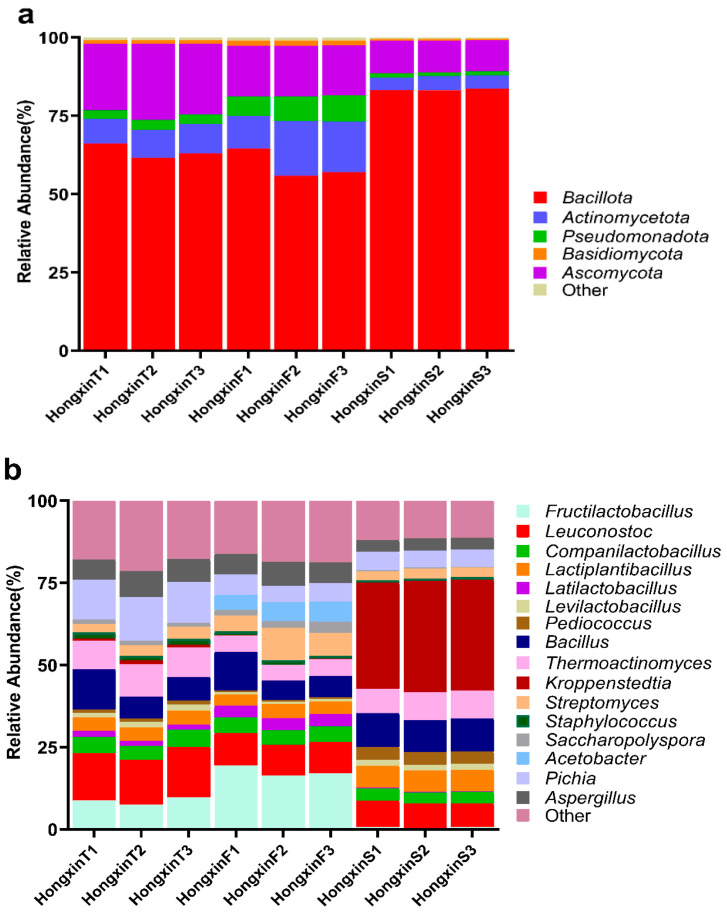
Relative abundance of microbiota at the phylum (**a**), genus (**b**), and species (**c**) levels for three classes of HX samples. Venn diagrams show common (**d**) and unique (**e**) microbial species in high-quality, primary, and secondary HX samples, with core species being those present in all samples. In Figures (**d**,**e**), the letter codes T, F, and S stand for top-grade HX, first-grade HX, and second-grade HX, respectively, and the suffix numeric codes indicate the sample number.

**Figure 3 foods-14-01104-f003:**
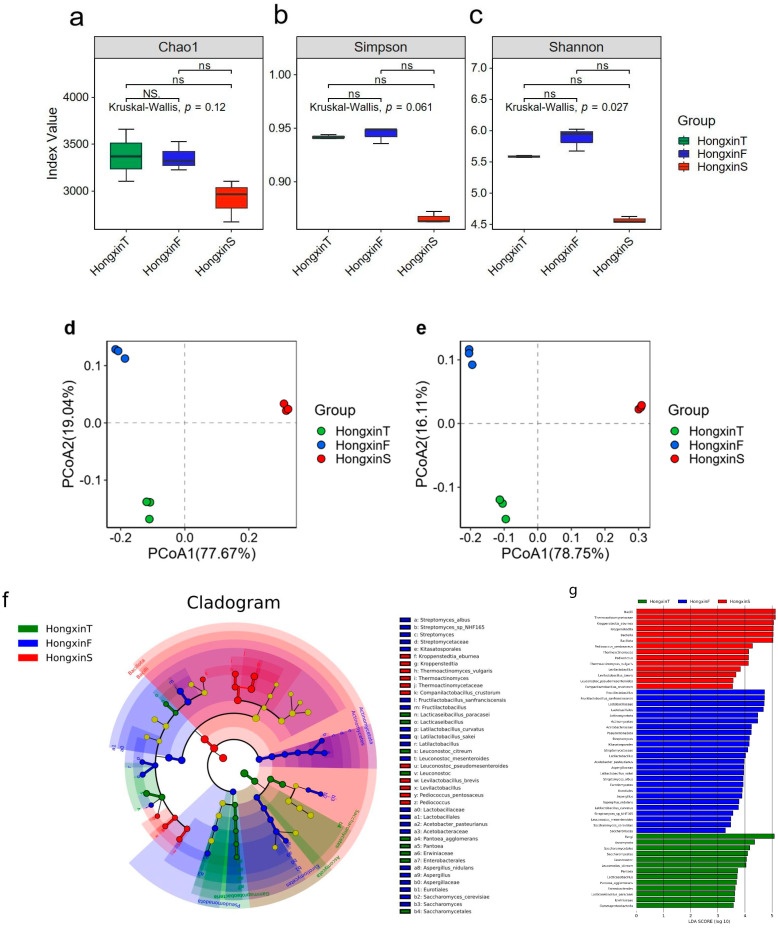
Comparative analysis of microbial community structure of HXT, HXF, and HXS with box line plots of microbiota α-diversity indices associated with the three grades of HX, including the Chao1 index (**a**), the Shannon diversity index (**b**), and the Simpson diversity index (**c**). Comparative analyses of microbial community structure for the three grades of HX: Binary Jaccard was utilized for plotting microbial β-diversity scores using binary Jaccard distance; (**d**) Jensen–Shannon divergence distance (**e**) to plot the principal coordinate analysis (PCoA) scores of microbial β-diversity; and linear discriminant analysis (LEfSe) was used to identify discriminatory microbial taxa of three different grades of HX. On the left is a branching diagram (**f**) showing the microbiota generated by LEfSe, and on the right is a horizontal bar graph (**g**) showing the discriminant taxa. Species with no differences are classified with yellow points, significantly different taxa are colored by group, and the color of the fan indicates a high-level species classification. The meaning of the LDA score is the contribution of the species to the differences between groups.

**Figure 4 foods-14-01104-f004:**
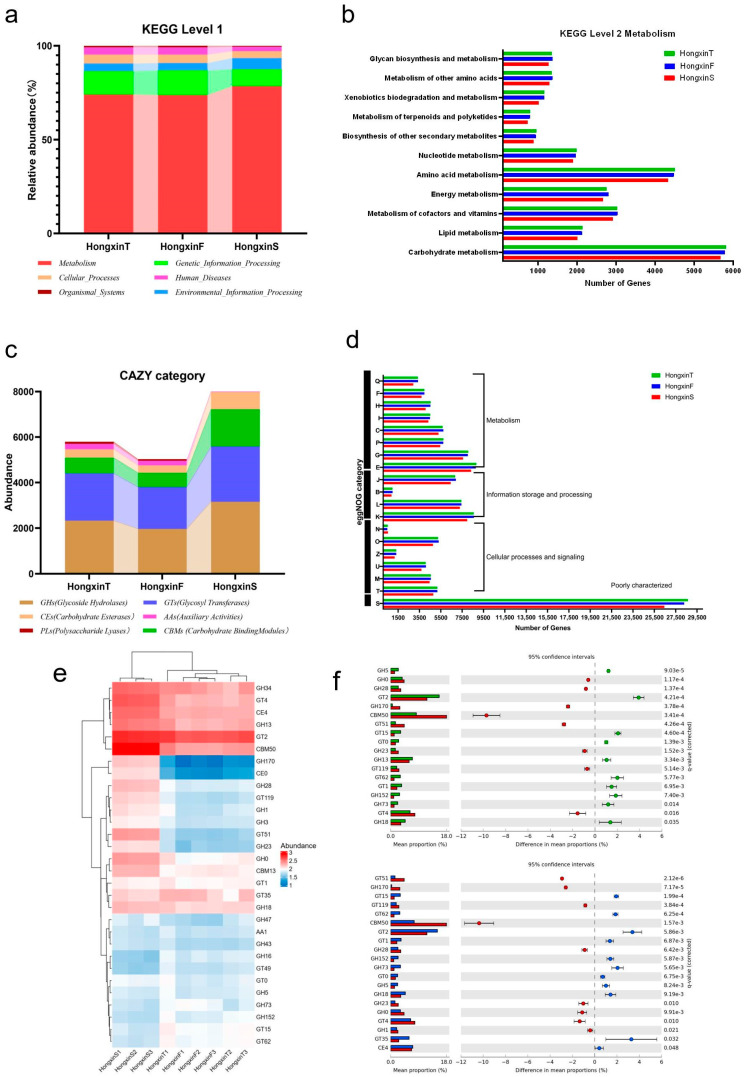
Functional annotation of HX bacteriophage genes from KEGG (**a**,**b**), CAZy (**c**), and eggNOG (**d**) databases reveals the variation in functional diversity among the three grades of HX-type bacteria. Heatmap showing the significant differential abundance between the three grades of HX relative abundance of CAZy family (**e**). The color scale in the figure (**f**) indicates relative abundance on a logarithmic scale ranging from high relative abundance (indicated in red) to low relative abundance (indicated in blue). The relative abundance of the carbohydrate-activating enzyme (CAZy) family differs significantly between HXS (in red), HXT (in green), and HXF (in blue) (**f**). The specific function description of each eggNOG category is shown in Appendix A.

**Figure 5 foods-14-01104-f005:**
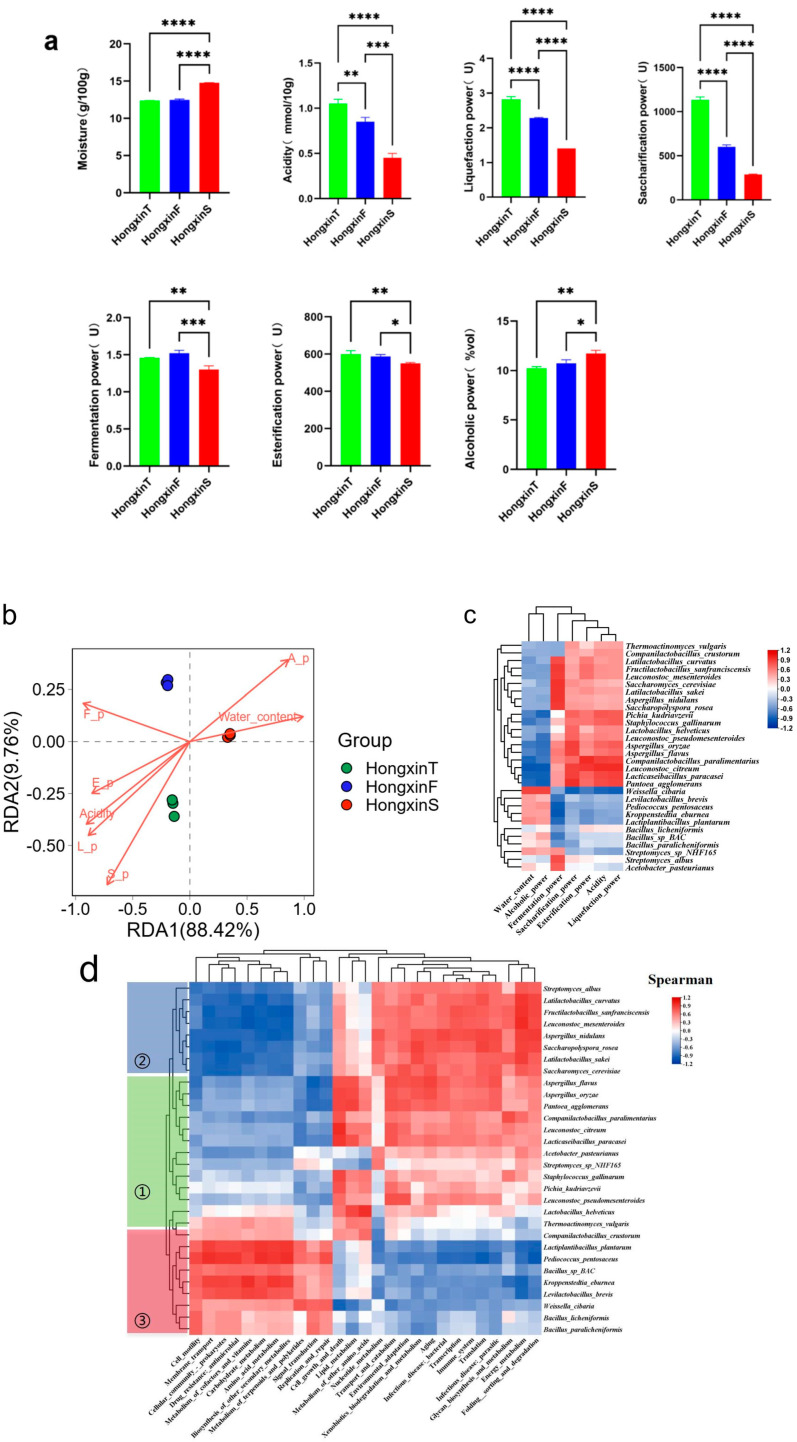
Analysis of the consistency and correlation between physical and chemical indicators and dominant species and functions. The histogram (**a**) depicts the content and difference in each physical and chemical index in three kinds of HX. Significant differences were expressed as **** (*p* < 0.0001), *** (*p* < 0.001), ** (0.001 ≤ *p* < 0.01), and * (0.01 ≤ *p* < 0.05). Redundancy analysis (RDA) was performed to determine the correlation between dominant species and the HX physicochemical index (**b**). Spearman correlation analysis was performed to determine the correlation between dominant species and physicochemical indicators (**c**) and microbial function (**d**).

## Data Availability

The original contributions presented in the study are included in the article/Appendix A; further inquiries can be directed to the corresponding author.

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
