# Peer review of "Metagenomic and Physicochemical Analyses Reveal Microbial Community and Functional Differences Between Three Different Grades of Hongxin Low-Temperature Daqu"

_foods, 2025, doi:10.3390/foods14071104_

Round 1

Reviewer 1 Report

Comments and Suggestions for Authors

The manuscript “Metagenomic and physicochemical analyses reveal microbial community and functional differences between three different grades of Hongxin low-temperature Daqu” presents a thorough study where metagenomics and physicochemical analysis were used to compare three grades of Hongxin daqu.

The results concerning the microbial population, the metagenomic functional characteristics and the physicochemical analysis highly improves the knowledge on their interactions and composition in different grade daqu samples.

The manuscript is generally well written, although mainly lines 385-390 and 417-420 need improvement. There are some names in the figures and figure captions that have low quality and need improvement.

Other comments that need revision are in the file attached.

Comments on the Quality of English Language

The manuscript is generally well written, although mainly lines 385-390 and 417-420 need improvement. 

Reviewer 2 Report

Comments and Suggestions for Authors

The present study demonstrates the application of a metagenomics approach for characterizing the microbial population in three different grades of Traditional Chinese Hongxin liquor. The metagenomic data were analyzed, compared, and correlated with changes in physicochemical characteristics during product fermentation. The manuscript was adequately prepared, but a more comprehensive version with an extensive discussion should be developed. In my humble opinion, several points as indicated below should be considerably clarified and revised.

  • Abstract: The abstract should be re-checked again according to the collective comments below. Please revise and clarify line 21-23.
  • Line 36: The abbreviation LFB should be clarified again here.
  • Line 55-81: This paragraph should be shortened to focus on the research question or the gap of knowledge addressed in this work.
  • Line 112: I suggest placing Figure 1 in Section 2.1, along with a more detailed description of the differences among the three grades of HX samples.
  • Line 122-126: References to these methods should be mentioned in the list of references.
  • Line 253: The resolution quality of Figure 3F and 3G should be improved.
  • Line 308: The clarity of labels in all Figure 4 panels should be improved.
  • Line 360: The clarity of labels in Figure 5C and 5D should be improved.
  • Line 390: The term “be killed” should be replaced by “be inhibited”.
  • Line 410-422: The authors should explain why the roles of these species are highlighted here.
  • Lione 449-512: The authors should discuss how the differences in parameters observed in this study could impact the final product quality. Additionally, the authors should explain how this information could contribute to optimizing the fermentation process to achieve the desired product quality, potentially with authentic characteristics. How the industrial sector could benefit from this information?

Reviewer 3 Report

Comments and Suggestions for Authors

The paper entitled “Metagenomic and physicochemical analyses reveal microbial community and functional differences between three different grades of Hongxin low-temperature Daqu” is very interesting research; however, it is necessary to adjust throughout the text:

  1. The introduction section is very long and includes 22 references out of 37 (59 %), so there must be a balance. Select the most relevant information that reflects the content of the article and contextualizes the need for metagenomic analysis.
  2. Figure 3. It is suggested to edit so that all elements are visible. Same case for fig 4 and 5 c and d.
  3. Line 407, citation format.
  4. Line 369 to 409. The references used to discuss are scarce, consequently the discussion is superficial, please use more references and strengthen the discussion.
  5. The conclusion should be aligned to what was stated in the last paragraph of the introduction or state the objective more specifically and align the conclusion to it.

Round 2

Reviewer 2 Report

Comments and Suggestions for Authors

Thank you for taking my comments into account.

I appreciate the authors' efforts in responding to the comments.